# Golgi Dysfunctions in Ciliopathies

**DOI:** 10.3390/cells11182773

**Published:** 2022-09-06

**Authors:** Justine Masson, Vincent El Ghouzzi

**Affiliations:** 1UMR-S 1270 Institut du Fer à Moulin INSERM, Sorbonne University, F-75005 Paris, France; 2NeuroDiderot, Inserm, Université Paris Cité, F-75019 Paris, France

**Keywords:** Golgi, primary cilium, ciliopathy, genetic screening, Golgipathies

## Abstract

The Golgi apparatus (GA) is essential for intracellular sorting, trafficking and the targeting of proteins to specific cellular compartments. Anatomically, the GA spreads all over the cell but is also particularly enriched close to the base of the primary cilium. This peculiar organelle protrudes at the surface of almost all cells and fulfills many cellular functions, in particular during development, when a dysfunction of the primary cilium can lead to disorders called ciliopathies. While ciliopathies caused by loss of ciliated proteins have been extensively documented, several studies suggest that alterations of GA and GA-associated proteins can also affect ciliogenesis. Here, we aim to discuss how the loss-of-function of genes coding these proteins induces ciliary defects and results in ciliopathies.

## 1. Introduction

The Golgi apparatus (GA) was described in 1898 by Camillo Golgi as an intracellular compartment that turns black when it is processed with a silver osmium technique coloration. This internal reticular apparatus was first observed in neurons, where it plays, as in all cells, a key role in the intracellular sorting, trafficking and targeting of proteins and lipids. GA dysfunction is involved in a wide range of diseases. However, whether it is a cause or a consequence of the disease is always a question. Considering a step further, when Golgi alterations/dysfunction is the cause, the induced cellular defects that are responsible for a clinical feature of the disease can be diverse (targeting to the cell membrane, glycosylation, etc.). Here, we will focus on one specific downstream effect of GA dysfunction that has been less explored, the alteration of the primary cilium, a peculiar thin organelle protruding at the cell surface. The primary cilium is composed of an axoneme made of nine pairs of microtubules that emerge from the basal body, which is derived from the mother centriole of the centrosome (Figure 1A). Most cells form a primary cilium when they are not dividing. Primary cilia regulate essential cellular processes, such as proliferation, differentiation and cell migration. Their dysfunction causes multiple organ diseases known as ciliopathies. In these pathologies, defects could affect immotile primary cilia but also motile cilia that carry, in addition to the nine outer doublets, two central microtubule singlets. Motile cilia are present in specialized cells in the respiratory epithelial cells, ependymal cells present on the ventricular surface of the brain, and in the fallopian tubes. While initially identified genes all encode proteins localized at cilia and centrosomes, it has been recently observed that genetic deficiency of proteins associated with other sub-compartments can also induce defects in ciliogenesis and ciliary signaling [1]. This new class of diseases, which are named second-order ciliopathies, is in particular associated with genes encoding for resident Golgi proteins or involved in Golgi maintenance. The aim of this review was to point out the importance of Golgi and post-Golgi trafficking in ciliary function and to discuss how a number of defects observed in diseases associated with mutations in Golgi proteins—referred to as Golgipathies—could be attributed to cilium dysfunction. We will focus on primary cilium, as clinical manifestations of motile ciliopathy [2] do not overlap with those of Golgipathies. We will first describe the anatomical relationship between the GA and the cilium and how pathologies that affect the GA or the primary cilium display common clinical features. We will then review Golgi proteins (resident and involved in the maintenance of this compartment) that were identified by the genomic screening of patients with ciliopathy (Table 1) or that were described as essential for ciliogenesis in animal (Table 2) and cellular models. As dividing cells do not form a cilium, the investigation of ciliogenesis in epithelial cell culture implies a serum starvation that reduces cell survival, increases autophagy, affects trafficking of proteins and certainly affects other essential cellular processes. Aware of these limitations, we have chosen to focus as far as possible on studies using animal tissues.

## 2. Golgi Apparatus and Primary Cilium Are Closely Linked

Primary cilia are immotile structures that are present at the surface of most cells. They are also named sensory cilia since they are sensors of external cues and transduce signals from the environment by expressing G-protein coupled receptors, receptor tyrosine kinases and receptors for morphogens [21]. Specialized immotile cilia named connecting cilia are observed in photoreceptors in the retina. They do not extend to sense extracellular cues but connect the inner and the outer segments of photoreceptors. The inner segment is the cell body that contains all subcellular organelles, including the GA; the outer segment contains rhodopsin that transduces photoexcitation (Figure 1B). As rhodopsin is degraded after stimulation, its traffic from the inner to the outer segment through the connecting cilium is essential for visual transduction. Whether a cell extends a primary cilium or a connecting cilium, immunolabeling or electron microcopy experiments have systematically shown the presence of GA saccules at the base of the cilium. The first evidence was provided in 1985 by Poole and al., who examined more than 300 primary cilia by electron microscopy in embryonic and mature connective tissue cells. This study showed that the basal body of the primary cilium was consistently associated with the maturing face of the GA complex and GA vacuoles [22]. Since then, this structural association between the primary cilium and the GA has been observed by electron microscopy in many other cells, such as rat pancreatic islet cells [23], rabbit epithelial cells, stromal fibroblasts and ciliary processes of the rabbit eye [24]. Published electron microscopy pictures of primary cilia quite often reveal a proximity between the basal body and the GA, as shown in Figure 2 (unpublished and kindly provided by Jorge Diaz, Université Paris Cité, CNRS, Integrative Neuroscience and Cognition Center, F-75006 Paris, France), in the mouse adult hippocampus. This proximity was demonstrated by double immunofluorescence experiments on ciliated cells [25,26].

Moreover, common clinical features of pathologies involving mutations in cilium-related and Golgi-related genes are observed. Surprisingly, despite a wide range of tissues affected and a high heterogeneity in clinical features, both classes of diseases, respectively referred to as ciliopathies [27] and Golgipathies [28], frequently affect the central nervous system, the retina, the skeleton and gonads (Figure 3).

## 3. Targeting Proteins to the Cilium

The GA is essential to process proteins and sort them to their functional destinations. This is emphasized, for example, in epithelial cells that have a basal and an apical compartment, or in neurons in which the post-Golgi sorting of vesicles will target some proteins to dendrites, whereas others will be confined in the soma or sent to the axon. In the majority of cells, the GA appears as a coherent and dynamic structure located in the soma, close to the nucleus, generally around the centriole. However, in specialized cells such as neurons that extend dendrites and axons, the GA is also detectable away from the nucleus, in particular in a number of dendrites where it appears as more discrete structures named Golgi outposts. Whether the GA associated with the basal body of ciliated cells represents a specialized Golgi sub-compartment has never been investigated, but these Golgi saccules could be either pericentriolar or far from the nucleus, as shown in Figure 3. How specific proteins are targeted to the primary cilium has been studied, and for transmembrane proteins, the first ciliary targeting motif identified was QVS(A)PA. In rhodopsin, the C terminus domain that lacks this sequence is blocked in the trans-Golgi network (TGN) and fails to reach the connecting cilium of photoreceptors [29]. This motif was later extended to VxPx, Ax[S/A]xQ, [I/V]KARK and RDYR [30,31,32,33,34], suggesting the existence of multiple targeting pathways. Through its ability to sort proteins and target them to the basal body or the cilium, the GA plays a critical role in ciliogenesis and in cilium signaling, which relies on the ciliary targeting of receptors that sense external cues and the presence of secondary messengers in the axoneme. In this context, the dysfunction of resident GA proteins, or of proteins involved in post-Golgi transport, or even of some proteins required for Golgi integrity but not directly located at the GA, could be responsible for ciliary defects observed in Golgipathies.

## 4. GA Related Proteins Necessary for Normal Ciliogenesis/Ciliary Function

### 4.1. Resident GA Proteins

#### 4.1.1. GIANTIN

GIANTIN was identified in 1993 as an integral component of the GA membrane with a disulfide-linked lumenal domain [35]. GIANTIN belongs to the golgin family that extends out from the surface of the GA to tether transport vesicles and other Golgi membranes.

GIANTIN is involved in glycosylation, and GIANTIN loss accelerates anterograde transport without affecting GA morphology [36,37,38], although it may inhibit lateral tethering between cisternae. Mutations in *Golgb1,* the gene encoding GIANTIN, or loss-of-function models display morphological alterations reminiscent of ciliopathies. In the rat, a spontaneous 10 bp insertion in *Golgb1* induces craniofacial abnormalities, edema, shorter limbs and embryonic lethality [11]. In knock out (KO) mice, the phenotype was mild with a cleft palate [12]. The link between GIANTIN and the primary cilium has been finally investigated in the zebrafish using morpholino and KO [39]. The acute knock down (KD) (embryos injected at ATG or E14 MO) displays characteristic phenotypes of ciliopathies in zebrafish larvae, such as small eyes and body length, kinks and curls in the tail, frequent edema of the heart and brain, and heart laterality defects. Using acetylated tubulin and Arl13B staining, this study further showed that GIANTIN morphants have less and longer cilia in the ependymal cells that border the ventral neural tube. These ciliary defects were confirmed by scanning electron microscopy in the olfactory placode, which bears kinocilia displaying a bulbous appearance of the ciliary tips, a phenotype reproduced in the LLC-PK1 cell line transfected with short interfering RNAs (siRNA) targeting GIANTIN. This phenotype was also observed in the KO zebrafish, but without the anatomical ciliary defects [39].

#### 4.1.2. GALNT11

GALNT11 is an N-acetylgalactosaminyltransferase whose role is to initiate the GalNAc-type O-glycosylation of proteins. Although GALNT11 subcellular localization has not been reported, it belongs to a class of enzymes acting at the GA and distributed throughout the Golgi stacks [40]. A rare genic deletion involving the *GALNT11* gene was identified in a patient with a congenital heart disease called heterotaxy and resulting from abnormalities in left–right body patterning, a landmark of ciliopathy [41]. *Xenopus* morpholinos targeting *galnt11* recapitulated cardiac defects observed in the patient that were rescued following overexpression of the human *GALNT11* transcript [42]. Using embryonic left–right organizer (LRO) explants that carry motile cilia in their center and primary cilia at the periphery, the authors showed that *galnt11* modulates the spatial distribution and ratio of motile and immotile cilia through Notch signaling activation. *Galnt11* loss-of-function thus impairs a balance between motile and immotile cilia essential to determine laterality at the LRO [42].

#### 4.1.3. IFT20 and Its Associated Complex

Ciliogenesis and ciliary signaling pathways rely on intraflagellar transport (IFT), which is performed by a class of molecules that contain binding sites for cargos and promote the retrograde and anterograde transport along the cilium axoneme. IFT proteins are expressed in the axoneme, and most of them are also expressed in the peri-basal body, except for IFT20 that is expressed in the cilium and at the GA. Using heterologous cellular models, IFT20 has been observed all along a thin thread extending from the GA to the base of the cilium. Whereas a full depletion of IFT20 prevents ciliogenesis, a weaker reduction only reduces the amount of the ciliary protein PC2 in the axoneme [43]. Electron microscopy analyses in photoreceptors also identified IFT20 at both the GA and the base of the cilium but not in the axoneme. Moreover, the depletion of IFT20 leads to opsin accumulation in the inner segment with a high concentration at the Golgi, without affecting the connecting cilium and outer segment morphologies [15]. Finally, in the male reproductive system, primary cilium differentiates into the sperm tail, and IFT20 has been shown to be essential in this process by transporting two flagellar proteins, ODF2 and SPAG16L [16].

Several proteins interacting with IFT20 have been identified. IFT20 is anchored to the GA by GMAP210/Trip11, and CCDC41 recruits IFT20 at the GA to target it to the basal body.

GMAP210/Trip11 is associated with microtubules, and its overexpression in heterologous cells (HeLa and COS-7) induces a dramatic enlargement of GA stacks and perturbates the microtubule network [44]. Oppositely, the deletion of *Trip11* affects the formation of the acrosome of sperm, a Golgi vesicle-derived structure [18], but no effect on GA structure was observed in mouse embryonic kidney (MEK) cells and fibroblasts [17]. Interestingly, MEK *Trip11*-depleted cells no longer express IFT20 at the GA and display a decrease in cilium length and a reduction in the expression of the ciliary protein PC2 [17], whereas flagella formation was not affected by the genetic deletion of *Trip11* [18]. In a mutant for *sql-1*, the *Caenorhabditis*
*elegans* homologue of Trip11, the Golgi is disorganized and the stability of the IFT machinery is affected, although the ciliogenesis does not seem to be disturbed, judging by a normal length of cilia [45]. Patients carrying hypomorphic mutations in the *TRIP11* gene display odontochondrodysplasia, a pathology affecting skeletal and dental development [3]. The functional analysis using fibroblasts isolated from skin patients demonstrated that the Golgi structure (evaluated by electron microscopy and GM130 and ZFPL1 staining) is altered with an increase in large vesicular elements. However, no changes in ciliary length (evaluated by acetylated tubulin) and ciliary expression of IFT20 were observed, which may be attributed to a residual variant GMAP210 expression in the cis-Golgi. Conversely, in fibroblasts isolated from fetuses carrying biallelic loss of *TRIP11*, the GA appeared highly compacted, the number of ciliated cells was drastically reduced (80%), and the few remaining cilia were reduced in length, as shown by the quantification of Arl13B and acetylated α tubulin [4]. The biallelic loss of *TRIP11* is lethal during embryonic life, and fetuses displayed achondrogenesis type 1A, a rare skeletal dysplasia [4,46]. How IFT20 depletion or mistargeting from the GA to the primary cilium affects ciliogenesis has been investigated by [47]. The docking of a GA-derived vesicle to the distal end of the mother centriole establishes a membrane–centriole association at the onset of ciliogenesis.

CCDC41/CEP83 was identified by centrosome proteomics as a distal appendage associated protein (DAP). This coiled-coil domain containing 41 proteins belongs to a DAP complex, and its loss leads to the failure of the undocked mother centriole to dock to the membrane, an essential step of ciliogenesis in RPE-1 cells [47,48]. A pool of CCDC41 colocalizes with GIANTIN and with IFT20, and the interaction of IFT20 with CCDC41 was confirmed by immunoprecipitation. Depletion of IFT20 induces a decrease in CCDC41 expression at the GA, whereas depletion of CCDC41 affects centrosomal IFT20 expression, suggesting that IFT20 plays a role in CCDC41 recruitment to the GA and that CCDC41 is involved in IFT20 recruitment to the centrosome [47].

#### 4.1.4. VPS15/PIK3R4

Vacuolar protein sorting 15 (VPS15) encodes the regulatory subunit 4 of the phosphoinositide 3-kinase (PI3K) complex required for the synthesis of the lipid phosphatidylinositol 3-phosphate. In the yeast *Saccharomyces cerevisiae*, VPS15 is required for vacuolar protein sorting [49], and in mammals, it is involved in autophagy [50]. The GA localization of VPS15 has been shown in a study in which the function of VPS15 in ciliogenesis was identified [5]. In this study, the cis-Golgi protein GM130 was found as an interactor of VPS15 in cell lysates from control human fibroblasts. Confocal microscopy further confirmed that a pool of VPS15 colocalizes with GM130. Patients carrying a missense mutation in the *VPS15* gene with clinical features compatible with a ciliopathy (early-onset retinal degeneration, late childhood kidney failure, mild skeletal developmental features and intellectual disability) were identified, and cells from these patients displayed shorter cilia. The interaction of VPS15 with GM130 was found unaffected. Since IFT20 is required for ciliary assembly and is anchored at the cis-Golgi, the localization of IFT20 at the GA was investigated upon cilium induction using serum starvation. While IFT20 was colocalized both at the cis-Golgi and scattered in vesicles throughout the cytoplasm in complete medium, serum starvation resulted in a restricted distribution of the protein in the GA, only in VPS15-deficient patient cells, suggesting a defect in the release of IFT20 positive vesicles from the cis-Golgi to the primary cilium [5]. The interplay between ciliogenesis and autophagy has been pointed out [51]. As in Vps15-depleted cells, autophagy flux is compromised (increase in the accumulation of adaptor protein p62, LC3 and Lamp2 positive vesicles); we can thus hypothesize that the loss of VPS15 could affect ciliogenesis through an abnormal autophagic process [50].

#### 4.1.5. KIF1C

Kinesin family members are motor proteins mediating the intracellular transport of various cargoes and protein complexes in a microtubule-dependent manner. KIF1C has been shown to associate to the minus end of microtubules via its tail domain and to serve as a linker between the GA and the microtubules to regulate Golgi positioning around the centrosome [52]. Like VPS15, KIF1C partially associates with GM130, and KIF1C depletion using CRISPR/Cas9 in HEK293T cells resulted in GA dispersal into small vesicles. Interestingly, KIF1C is also required to export membrane proteins from the GA to the primary cilium, as its knockdown severely inhibited serum starvation-induced ciliogenesis in RPE1 cells, resulting in the accumulation of ciliary receptors in the Golgi area [53]. In particular, the recruitment of IFT20 to the basal body of the cilium was markedly reduced in KIF1C-depleted cells. KIF1C-mediated transport to the cilium was proposed to involve a physical interaction with ASAP1, an ARF GTPase-activating protein serving as a scaffold protein to regulate ARF4, RAB8/RAB11-mediated ciliary receptor targeting [54]. Several studies have identified patients carrying mutations in the *KIF1C* gene as suffering from hereditary spastic paraparesis, a disorder characterized by spasticity predominantly in the lower limb that could be associated with cerebellar ataxia and upper cervical atrophy [6,7,8,55]. None of these studies investigated potential defects in ciliogenesis. However, it is interesting to note that spasticity and cerebellar defects are often observed in patients with ciliopathies.

### 4.2. Golgi Proteins Involved in Post-Golgi Transport

ARF4 is a small G protein belonging to the Ras-related subfamily proteins. It is localized primarily to the GA and is essential for the secretory pathway. In photoreceptors, ARF4 binds to rhodopsin thanks to the VxPx ciliary motif [31]. It regulates the rhodopsin association with the TGN and with a protein complex composed of ARF4, ASAP1, FIP3 and RAB8/RAB11 that controls the budding of post-TGN carriers to target rhodopsin to the primary cilium [56]. To dissect the mechanism responsible for the targeting of the ARF4 complex to the cilium, a cellular model of the mouse inner medullary collecting duct cells (IMDC3) was used, as it is able to build a cilium and to target rhodopsin into the axoneme. Depleting ASAP1 (known as an actin regulator) in IMDC3 abolished the targeting of rhodopsin to the cilium and accumulated it in actin-rich juxtaciliary protrusions. In this context, ASAP1 seems to be involved in a post-Golgi step allowing the translocation across the periciliary diffusion barrier [54]. However, in photoreceptors, post-TGN rhodopsin ciliary-targeted carriers (RTC) are generated, and it was shown using sucrose gradient experiments that while ASAP1 and RAB11 co-fractionated with GA/TGN and RTC markers, ARF4 only associated with GA/TGN compartments. This shows that ARF4 is involved in the progression out of the TGN before the release of RTC from the TGN, whereas ASAP1 acts as a scaffold for RAB8/11 for the budding of RTC [54].

### 4.3. Proteins Required for Golgi Integrity but Not Directly Located at the GA

Morphological integrity and functionality of the GA both depend on anterograde and retrograde trafficking with the endoplasmic reticulum through the intermediate/ERGIC compartment [57]. The GA and the centrosome (the major microtubule organizing center required for ciliogenesis) have close physical proximity and are functionally linked by a microtubule network (Figure 1). How disrupting Golgi–centrosome proximity can affect ciliogenesis is still debated [58], but one can hypothesize that centrosomal protein deletion affects GA structure, that in turn will reinforce ciliary defects primarily induced by centrosomal alterations.

#### 4.3.1. YIF1B

Recently, Diaz et al. demonstrated that a protein expressed in the intermediate compartment between the ER and the GA is required for normal ciliogenesis [9]. The YIF1B protein was initially isolated as a chaperone protein for the targeting of the serotonergic receptor in the dendritic arborization of neurons [59]. Subcellular expression of YIF1B in heterologous cells and neuronal cultures and pharmacological treatments disturbing anterograde and retrograde transport from the ER to the cell membrane demonstrated that YIF1B is not a resident protein of the GA but is rather located in the vesicles trafficking between the ER and the GA [19]. Constitutive loss of YIF1B induces GA fragmentation in mouse neurons and defects in cilium/basal body structure in skin fibroblasts from both KO mice and patients carrying mutations in the *YIF1B* gene [9].

#### 4.3.2. TAPT1

In human fibroblasts, TAPT1 is localized at the basal body, and its loss leads to a dispersion of the GA [10]. TAPT1 is a transmembrane anterior posterior transformation 1 protein (TAPT1) identified as an effector of HOXC8, a homeobox protein involved in the antero-posterior developmental patterning in mice [20]. It is ubiquitously expressed, and while its function remains unknown, it has been shown to be required for axial skeletal patterning during development. Mutations in the human *TAPT1* gene cause a complex lethal osteochondrodysplasia and disrupt ciliogenesis [10]. Using a zebrafish morpholino, the depletion of TAPT1P, the zebrafish TAPT1 homologue, induces a decrease in ciliary length. Unfortunately, this study did not investigate the GA structure in the zebrafish model, although the skeletal malformations described resemble those of affected patients [10]. TBCCD1, the TBCC-domain containing 1 protein related to tubulin cofactor C, localizes at the centrosome. Although no mutations in the *TBCCD1* gene have been identified in patients displaying ciliopathic-like symptoms so far, its depletion caused a disorganization of the GA and a decrease in the percentage of RPE-1 cells bearing a primary cilium after serum deprivation [60], suggesting, as for TAPT1, that a centrosomal protein loss may indirectly affect ciliogenesis through the disorganization of the GA.

## 5. Conclusions

In 2013, the SYSCILIA consortium proposed a list of genes (SCGSv.1, http://www.syscilia.org/goldstandard.shtml (accessed on 5 September 2022)) related to ciliary function, and several of them encode proteins expressed in the GA. In 2017, Reiter and Leroux proposed to group ciliopathies caused by the loss of extra-ciliary proteins under the name of second-order ciliopathies, and this group obviously includes GA-related ciliopathies. So far, genomic DNA from patients with a clinical suspicion of ciliopathy are tested with ciliopathy panels only consisting of ciliary genes (https://www.ncbi.nlm.nih.gov/gtr/tests/552623/overview/ (accessed on 5 September 2022)), and in case of the absence of the identification of a mutation in these genes, whole genome sequencing is performed. The close functional relationship between GA and primary cilium suggests that genes coding for GA proteins should be considered in a second intention.

## Figures and Tables

**Figure 1 cells-11-02773-f001:**
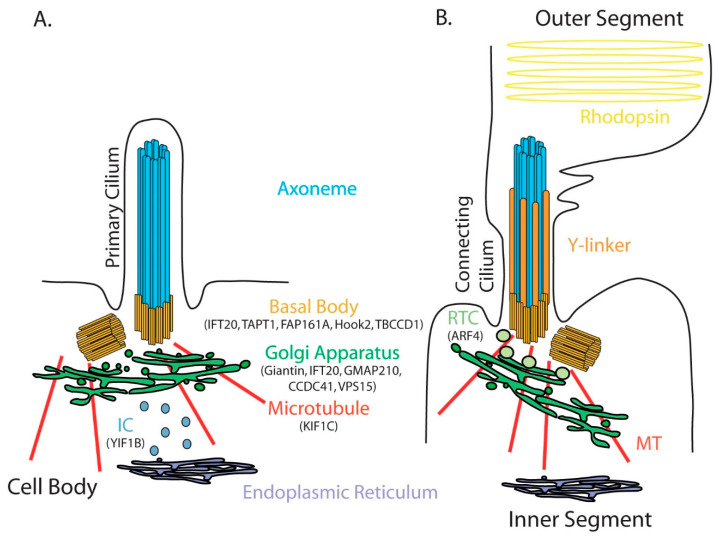
Schematic representation of the primary (**A**) and connecting cilium (**B**). The axoneme contains nine pairs of microtubules grown from the mother centriole differentiated into a basal body. The Golgi apparatus is anatomically closely related to the cilium. In photoreceptors, the connecting cilium links the photoreceptor inner and outer segments and has a specialized transition zone composed of Y-linkers and specialized post-Golgi rhodopsin carriers (RTC) that allow the targeting of rhodopsin to the outer segment. IC is the intermediate compartment between the endoplasmic reticulum and the Golgi apparatus. Microtubules allow dynein-dependent trafficking between the centrioles and the Golgi apparatus. All proteins mentioned in the manuscript as being involved in ciliogenesis/ciliary function appear in their resident compartment.

**Figure 2 cells-11-02773-f002:**
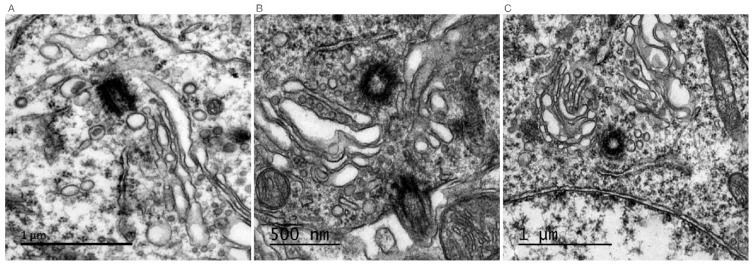
Electron microscopy pictures of hippocampal pyramidal cells in the mouse brain (unpublished and kindly provided by Jorge Diaz, Université Paris Cité, CNRS, Integrative Neuroscience and Cognition Center, F-75006 Paris, France). The basal body is visible in (**A**), the mother centriole in (**B**,**C**), and the axoneme of a primary cilium in (**B**). Whether the primary cilium extends close (**C**) or far (**B**) from the nucleus, Golgi apparatus saccules coat the base [6].

**Figure 3 cells-11-02773-f003:**
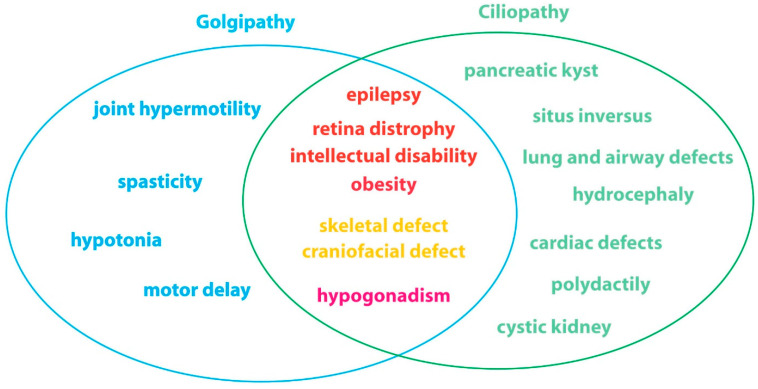
Clinical features of Golgipathies and ciliopathies. Whereas some defects are specific tor one disease class (in blue for Golgipathy and in green for ciliopathy), some are common and concern alterations in the central nervous system (in red, including obesity that has been described to be caused by hyperphagia), the skeleton (in orange) and the gonad (in pink).

**Table 1 cells-11-02773-t001:** List of mutated genes coding for GA and GA-associated proteins identified in patients presenting some of the clinical features of ciliopathies. *, stop codon.

Gene	Nucleotide Change	Predicted Amino Acid Change	Protein Level	Reference
*GMAP210/Trip11*	c.[1314+5G>A]; [chr14:g. (?_ 92.474.069)_ (92.597.431_?)del	p.[(Glu439Val*fs**20)]; [(?)]A	N.D	[3]
	c.[1228G>T]; [4815_4818delAGAG]	p.[Asp410_ Lys438del]; [Glu1606Leu*fs**3]	Strongly reduced (reduction of approximately 90%)	
	c.[586C>T];[4534C>T]	p.[Gln196*]; (Gln1512*]	N.D	
	c.[1228G>T]; [2128_2129delAT]	p.[Asp410Tyr]; [Ile710Cys*fs**19]	Complete absence	
	c.[1622delA]; [5416A>G]	p.[Lys541Arg*fs**17]; [Met1806Val]	Complete absence	
	c.[586C>T]; [2993_2994delAA]	p.[Gln196*]; [(ys998Ser*fs**5]	N.D	
	c.[586C>T]; [790C>T]	p.[Gln196*]; [Arg264*]	Reduced (30% of control)	
	c.[5457+81T>A]	N.D	Complete absence	[4]
*VPS15*	c.[2993G>A]	p.[Arg998Gl]	N.D	[5]
*KIF1C*	c.[4925567C>T]	N.D	Complete absence	[6]
	c.[1019+1dup]	N.D	N.D	[7]
	c.[1166-2A>G]	N.D	N.D	[8]
*YIF1B*	c.[598G>T]	p.[Glu200*]	N.D	[9]
	c.[539+1G>A]	p.[Ala161Glyfs*18]	Complete absence	
	c.[367A>C]	p.[Lys123Gln]	Strongly reduced	
	c.[177dupT]	p.[Ala63Cysfs*13]	N.D	
	c.[539+1G>A]; [696-2A>C]	p.[Ala161Glyfs*18]; [Met233Phefs*40]	Complete absence	
	c.[577_592del); [501C>A]	p.[Ala193Profs*36]; [Tyr167*]	N.D	
*TAPT1*	c.[1108-1G>C]	p.[Val370_Asn389del]	N.D	[10]

**Table 2 cells-11-02773-t002:** List of mammalian animal models lacking expression of GA and GA-associated proteins.

Gene	Model	Silencing Strategy	Protein Level	Reference
*GIANTIN*	OCD rat (mutant inbred strain)	10-bp (CTGCAGGCAG) duplication in the 13th exon resulting in a frame shift mutation and introducing a premature termination codon at codon 1082	Absent	[11]
	Golgb1 KO mice (CRISPR/Cas9-mediated genome editing)	Extension of exon 9 introducing a premature termination codon	Absent	[12]
*IFT20*	Ift20 *^flox/flox^* [13]/HRGP-Cre mice [14]	Homologous recombination with a disrupting cassette	Absent in cone photoreceptor	[15]
	Ift20 *^flox/flox^* [13]/*Stra8-iCre* (Jackson Laboratory) mice	Homologous recombination with a disrupting cassette	Absent in male germ cell	[16]
*GMAP210/Trip11*	*Gmap210 KO mice*	Insertion of a splice acceptor site and a βgalactosidase-neomycin resistance gene fusion into intron 4 and duplication of Chr16	Absent	[17]
	*Gmap210^flox/flox^* [17]/*Stra8-iCre* (Jackson Laboratory) mice	Homologous recombination with a disrupting cassette	Absent in male germ cell	[18]
*YIF1B*	*Yif1B* KO mice [19]	Homologous recombination with a disrupting cassette	Absent	[9]
*TAPT1*	mice *carrying L5Jcs1* mutation	T>A in the 5′ splice site of intron 7 resulting in the introduction of 12 aa after N279 and a premature termination codon	N.D	[20]

## Data Availability

Not applicable.

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
