# Peer review of "Golgi Dysfunctions in Ciliopathies"

_cells, 2022, doi:10.3390/cells11182773_

Round 1
Reviewer 1 Report
The authors summarized the Golgi dysfunctions in ciliopathies. They first described the connection between the Golgi apparatus and primary cilium; then highlighted the conserved domains that mediate ciliary targeting; finally they listed the GA related proteins necessary for normal ciliogenesis/ciliary function.
The review is concise. I have two suggestions.
1. The authors concentrated on primary cilia, they should also discuss the motile cilia;
2. Increasing evidence showed that reproductive system is also affected. The authors should cite some of the findings, including the IFT20 and TRIP11 studies.
Author Response
We answered point-by-point to the reviewer’s comments (please see attachment).

Reviewer 2 Report
Golgi Apparatus is a subcellular compartment structured as a stack of cisternae designated to process and package proteins that have exited the ER. Eventually, the proteins can be either secreted in the extracellular environment or further transported inside the cell. Additionally, the Golgi Apparatus contributes to lysosome biogenesis. Hence, Golgi dysfunctions might be associated with a number of human disorders including, cancer, neurodegenerative-, infectious- and cardiovascular- diseases. The primary cilium is a structure composed by microtubules and quite a few ciliary proteins that are trafficked from the Golgi Apparatus to the base of the cilium and thereafter into the ciliary compartment. The primary cilium is implied in a number of cellular processes, including cell migration and motility. Primary cilia dysfunctions cause disorders, known as ciliophaties, that encompass most human organs. In the review article titled "Golgi dysfunctions in ciliophaties" the authors aim to discuss how the loss of function of Golgi apparatus-associated proteins induce ciliary defects, thus leading to ciliophathies. Prior to being considered for publication, the manuscript needs to be slightly amended and a few issues, below shortly discussed, need to be sorted out. Major concerns- The manuscript would benefit from a summary table in which, in case of mammals, there are listed the mutated genes alongside the corresponding nucleotide sequence, the position, the matching mutated aminoacid, and the respective references. Alternatively, if data are coming from gene silencing the approach used should be summarized;
- It appears that the loss of affecting the Golgi localized protein glycosylyl transferase/GALNT (Boskovsky M. T. et al Nature 2013) is associated with a kind of ciliary defect. Is there any reason why it has not been discussed?
- Few cell lines have been used as a model to investigate ciliopathies. The authors did not mention them. It would be greatly appreciated if the authors could shortly discuss the pros and cons of these tools;
- Since Golgi Apparatus significantly to the lysosomal biogenesis to the reviewer it is not clear why the mutated genes affect only cilia functionality but not the lysosomal one. Alternatively, is the lysosomal activity only mildly affected. This point should be clarified and articulated;
- HACE1 controls the Golgi Apparatus membranes dynamics during the cell cycle (Tang D. et al. Nat Comm. 2011). Is anything known about the ubiquitin system and ciliophaties? Based on the above-mentioned evidence the authors should discuss this matter;
Author Response
We answered point-by-point to the reviewer’s comments (see the attachment).

Round 2
Reviewer 2 Report
I thank the authors very much and I truly appreciate the efforts made to improve their review. Indeed, the revised version is now more comprehensive. Just a couple of minor things need to be further amended: a) when it comes to Figure 2 since Jorge Diaz is not an author it should be shortly indicated the affiliation (i.e. J. Diaz, University of ..../Institute .....); b) I still noticed that a few typos need to be edited. For examples see lines 205 (cerevisiae elegans???) and 235 (genus uppercase, species lowercase); c) The language used for tables 1 and 2 needs to be slightly edited. For example: "Strongly reduced (less that 10% of control)" should be "Strongly reduced (reduction of approximately 90%)"; "Complete absence in male germ cell specific", if the impaired expression is restricted only to male germ cells please it should be edited as follows "Absent in male germ cells", and few others more.Author Response
We thank the reviewer for his carreful reading of the revised manuscript. We have added Jorge Diaz's affiliation, corrected the two typos and made the changes in the table 2.